# Alteration of Gut Microbiota Composition in the Progression of Liver Damage in Patients with Metabolic Dysfunction-Associated Steatotic Liver Disease (MASLD)

**DOI:** 10.3390/ijms25084387

**Published:** 2024-04-16

**Authors:** Alejandra Zazueta, Lucía Valenzuela-Pérez, Nicolás Ortiz-López, Araceli Pinto-León, Verónica Torres, Danette Guiñez, Nicolás Aliaga, Pablo Merino, Alexandra Sandoval, Natalia Covarrubias, Edith Pérez de Arce, Máximo Cattaneo, Alvaro Urzúa, Juan Pablo Roblero, Jaime Poniachik, Martín Gotteland, Fabien Magne, Caroll Jenny Beltrán

**Affiliations:** 1Microbiology and Mycology Program, Institute of Biomedical Sciences, Faculty of Medicine, University of Chile, Santiago 8380453, Chile; alezazueta28@gmail.com; 2Laboratory of Immuno-Gastroenterology, Section of Gastroenterology, Department of Medicine, Hospital Clínico Universidad de Chile, Santiago 8380456, Chile; lucia.valenzuela.perez@gmail.com (L.V.-P.); nicolas.ortiz@ug.uchile.cl (N.O.-L.); aracelipleon@gmail.com (A.P.-L.); vf.torres.martinez@gmail.com (V.T.); nico.aliaga.t@gmail.com (N.A.); merinog.pablo@gmail.com (P.M.); 3Unit of Gastroenterology, Department of Medicine, Hospital Clinico Universidad de Chile, Santiago 8380456, Chile; dannette.vania@uchile.cl (D.G.); asandovalv@ug.uchile.cl (A.S.); ncovarrubias@hcuch.cl (N.C.); eperezdearce@gmail.com (E.P.d.A.); maximocattaneo@gmail.com (M.C.); dralvarourzua@gmail.com (A.U.); jproblero@gmail.com (J.P.R.); jponiachik@hcuch.cl (J.P.); 4Department of Nutrition, Faculty of Medicine, University of Chile, Santiago 8380453, Chile

**Keywords:** intestinal microbiota, MASLD, gut–liver axis, liver fibrosis, obesity

## Abstract

Metabolic dysfunction-associated steatotic liver disease (MASLD) is a complex disorder whose prevalence is rapidly growing in South America. The disturbances in the microbiota–gut–liver axis impact the liver damaging processes toward fibrosis. Gut microbiota status is shaped by dietary and lifestyle factors, depending on geographic location. We aimed to identify microbial signatures in a group of Chilean MASLD patients. Forty subjects were recruited, including healthy controls (HCs), overweight/obese subjects (Ow/Ob), patients with MASLD without fibrosis (MASLD/F−), and MASLD with fibrosis (MASLD/F+). Both MASLD and fibrosis were detected through elastography and/or biopsy, and fecal microbiota were analyzed through deep sequencing. Despite no differences in α- and β-diversity among all groups, a higher abundance of *Bilophila* and a lower presence of Defluviitaleaceae, Lachnospiraceae ND3007, and *Coprobacter* was found in MASLD/F− and MASLD/F+, compared to HC. Ruminococcaceae UCG-013 and *Sellimonas* were more abundant in MASLD/F+ than in Ow/Ob; both significantly differed between MASLD/F− and MASLD/F+, compared to HC. Significant positive correlations were observed between liver stiffness and *Bifidobacterium*, *Prevotella*, *Sarcina*, and *Acidaminococcus* abundance. Our results show that MASLD is associated with changes in bacterial taxa that are known to be involved in bile acid metabolism and SCFA production, with some of them being more specifically linked to fibrosis.

## 1. Introduction

Metabolic dysfunction-associated steatotic liver disease (MASLD), formerly known as non-alcoholic fatty liver disease (NAFLD), is a complex and heterogeneous disorder currently considered the hepatic manifestation of metabolic syndrome [1,2]. It is characterized by hepatic steatosis, affecting at least 5% of the biopsy specimens in subjects without heavy alcohol consumption or secondary causes of chronic liver disease such as viral- or drug-induced hepatitis. MASLD encompasses a broad spectrum of diseases, ranging from simple steatosis to non-alcoholic steatohepatitis (NASH), that can lead to liver fibrosis, cirrhosis, and hepatocellular carcinoma [3]. The global prevalence of MASLD has increased over the past decades, from 21.9% in 1991 to 37.3% in 2019, becoming a global public health problem [2]. Interestingly, South America has experienced the most rapid change in MASLD prevalence (2.7% per year), possibly due to racial and ethnic disparities as observed in the United States, where the most affected population is the Hispanic [4].

The pathogenesis of MASLD is complex and multifactorial. The initial event is generally considered to be the excessive accumulation of lipids in the liver because of the consumption of a high-calorie, high-fat diet, accompanied by increased adipose tissue lipolysis and de novo hepatic lipogenesis [1]. Other factors also participate in the pathophysiology of MASLD, including genetic and epigenetic factors, insulin resistance, lipotoxicity, oxidative stress, mitochondrial dysfunction, endoplasmic reticulum stress, and alterations in the microbiota–gut–liver axis [5,6]. Indeed, alterations in gut microbiota (GM) composition in MASLD patients have been associated with increased gut permeability, leading to the translocation of bacterial products from the gut to the liver via the enterohepatic circulation and contributing to the progression of liver injury, steatohepatitis, and fibrosis [7,8]. A relevant factor influencing GM composition is the geographical location of the subjects, which is associated with specific genetic, dietary, and lifestyle factors [9]. However, it is not known whether geography-related gut microbiota composition contributes to the higher risk or severity of MASLD, specifically in Latin America, where data are rather scarce. Therefore, the aim of this study was to identify microbial signatures in Chilean MASLD patients.

## 2. Results

### 2.1. Population Description

The subjects included in this pilot study were consecutively recruited from the hepatology or general gastroenterology consultation in the Hospital Clinico Universidad de Chile (HCUCH), according to inclusion and exclusion criteria, and after signing written informed consents. We included 7 healthy controls (HCs), 13 overweight/obese subjects (Ow/Ob), 11 patients with MASLD without fibrosis (MASLD/F−), and 9 MASLD patients with fibrosis (MASLD/F+). The MASLD diagnosis was performed through transient elastography (Fibroscan^®^) and/or biopsy histology. The demographic and clinical characteristics of the participants are summarized in Table 1. Compared with the HCs, MASLD/F+ patients were older (*p* = 0.026), and Ow/Ob subjects and MASLD/F+ patients had a higher weight (*p* = 0.0338 and *p* = 0.0027, respectively). The Ow/Ob subjects and the MASLD/F− and MASLD/F+ patients also had a higher BMI than the HCs (*p* = 0.0069, *p* = 0.0055, and *p* = 0.0003, respectively). Similarly, waist circumference was higher in MASLD/F− (*p* = 0.0377) and MASLD/F+ (*p* = 0.0073) patients, compared to the HCs. Regarding biochemical parameters, higher levels of AST (*p* = 0.0494) were observed in MASLD/F− patients than in the HCs. GGT levels were also higher in MASLD/F+ patients than in the HCs (*p* = 0.0093) and Ow/Ob subjects (*p* = 0.0302). Likewise, plasma cholesterol (*p* = 0.0220) and albumin (*p* = 0.0041) levels were higher in MASLD/F− patients than in MASLD/F+. The reported comorbidities included DM, hypertension, dyslipidemia, and atherosclerosis, which were principally observed in all the patients with MASLD.

### 2.2. Changes in Gut Microbiota in MASLD Patients

This study aimed to investigate whether MASLD was associated with changes in gut microbiota composition. For this purpose, a comparative analysis was performed between patients with MASLD (independently of the presence/absence of fibrosis) and HCs or Ow/Ob subjects. No significant differences between the groups were observed regarding the microbiota alpha-diversity (Figure 1A,B) and beta-diversity (Figure 2). Nevertheless, the CCA analysis clustered the gut microbiota samples into three groups, suggesting a possible association between MASLD and the gut microbiota. The comparison of microbial relative abundances between the different groups shows that several taxa were altered in the gut microbiota of MASLD patients, compared to both the HCs and Ow/Ob subjects (Figure 3). Compared to the HCs, MASLD patients had a higher abundance of *Bilophila* (*p* = 0.007) and a lower abundance of the Defluviitaleaceae family (*p* = 0.02), and Lachnospiraceae ND 3007 (*p* = 0.036) and *Coprobacter* (*p* = 0.032) genera. When the MASLD patients were compared to the Ow/Ob subjects, non-significant changes in the abundance of *Coprobacter* (*p* = 0.05) and Defluviitaleaceae were observed. In addition, the *Sutterella* genus was non-significantly increased in the MASLD patients compared to the Ow/Ob subjects (*p* = 0.07).

To further determine whether the severity of fibrosis in MASLD patients was associated with changes in microbial taxa, we compared the composition of the gut microbiota of the patients that were previously clustered into four groups according to their BMI index and severity of liver fibrosis: HCs (BMI < 25), Ow/Ob subjects (BMI > 25), MASLD/F−, and MASLD/F+. The abundance of the Ruminococcaceae UCG 013 and *Sellimonas* genera was significantly higher (*p* = 0.009) in MASLD/F+ patients compared to the Ow/Ob subjects. Interestingly, the abundance of these genera significantly differed between MASLD/F− and MASLD/F+ patients. Compared with the HCs, the relative abundance of the Ruminococcaceae UCG 013 and *Ruminiclostridium*_6 genera were lower (*p* = 0.003 and *p* = 0.023, respectively) in MASLD/F− and MASLD/F+ patients, whereas that of *Sellimonas* was higher (*p* = 0.019).

In addition, liver stiffness values positively correlated with the abundance of Actinobacteria phylum (R = 0.6, *p* = 0.0053) (Figure 4A), Bifidobacteriaceae family (R = 0.68, *p* = 0.00067) (Figure 4B), and *Bifidobacterium* (R = 0.68, *p* = 0.00072) (Figure 4C), *Prevotella* 2 (R = 0.85, *p* = 2.28 × 10^−9^) (Figure 4D), *Sarcina* (R = 0.85, *p* = 2.28 × 10^−9^) (Figure 4E), and *Acidaminococcus* genera (R = 0.63, *p* = 0.0063) (Figure 4F).

## 3. Discussion

Our results provide valuable insights into the interaction between the intestinal microbiota and the pathogenesis of MASLD. Certain bacterial taxa were effectively associated with MASLD and the severity of liver damage, suggesting that they could be used to predict MASLD progression. Nevertheless, some confounding variables, such as age and BMI, also influence gut microbiota composition, eventually masking the bacterial taxa predictor associated with MASLD progression [10,11]. Indeed, intestinal dysbiosis predisposes individuals to age-related diseases, among them MASLD [10], which an increased prevalence, particularly at an advanced stage, has been reported principally in older patients with this disease [12].

The pathophysiology of MASLD involves liver fat accumulation (steatosis) and inflammation that can lead to liver fibrosis. Although MASLD is a complex process, there is increasing evidence that gut microbiota contributes to liver fat accumulation and inflammation involved in liver fibrosis. In our study, various bacterial taxa were associated with MASLD. However, the comparative analysis of the MASLD patients with HCs (BMI < 25) and the Ow/Ob subjects (BMI > 25) present discrepancies. Only two taxa, the *Coprobacter* genus and the Defluviitaleaceae family, differed in their relative abundance in both analyses, although the difference was not significant for the second taxa. In our study, the MASLD patients had a BMI > 25, suggesting that the differentially abundant taxa identified between these patients and the HC could result mainly from the BMI difference. We considered, therefore, that the comparison between the MASLD patients and the Ow/Ob subjects was more appropriate since BMI was not statistically different between both groups, which could improve the identification of bacterial taxa associated with MASLD pathogenesis. Compared to the Ow/Ob subjects, MASLD patients presented an alteration in the abundance of *Coprobacter*, Defluviitaleaceae, and *Sutterella*.

These changes in the gut microbiota could be in part due to bile acid (BA) production, which is altered in MASLD patients. Indeed, studies have reported increases in the liver expression of genes involved in BA synthesis (CYP7B1) [13] and in BA serum levels in these patients, compared to healthy individuals [14]. BA is known to regulate the composition of the intestinal microbiota since low BA levels are associated with a lower number of Gram-positive bacteria [15]. In humans, the administration of obeticholic acid has been shown to increase the abundance of Gram-positive bacteria [16], while these effects in mice were reversed using BA sequestrants [17]. Together, these findings suggest that the changes observed in the gut microbiota of MASLD patients could be in part due to BA alteration that promotes bacterial death, although bacterial diversity did not decrease in our MASLD patients.

Furthermore, changes in some genera, such as Lachnospiraceae ND 3007, Ruminococcaceae UCG 013, and *Bilophila*, were also observed, suggesting an alteration in the formation of secondary BAs. Our MASLD patients had an increased proportion of *Bilophila* genus, known to limit the synthesis of secondary BAs, as reported with *B. wadsworthia* [18]. The Lachnospiraceae ND 3007 genus, belonging to the Lachnospiraceae family, was diminished in our MASLD patients, and some members of this family can convert primary BAs to secondary BAs in the colon. For example, certain species of *Blautia* can perform 7-α-dehydroxylation of primary BAs and convert them into secondary BAs such as lithocholic acid and deoxycholic acid [19]. In addition, the microbiota of our MASLD patients was enriched in Ruminococcaceae UCG 013; some members of the Ruminococcaceae family can produce ursodeoxycholic acid (UDCA), such as *Ruminococcus gnavus* [20]. Collectively, these data suggest that the MASLD patients exhibited changes in microorganisms that are involved in BA metabolism, suggesting alterations in the pool of secondary BAs.

The role of secondary BAs in the pathogenesis of MASLD is not clear since they have a paradoxical effect on the Farnesoid X Receptor (FXR), acting either as an agonist or antagonist with a distinct affinity according to their structure [21]. The FXR represents a relevant clinical target for MASLD treatment [22], since FXR activation has been shown to reduce hepatic and plasma levels of TAGs and liver inflammation [22], while increasing insulin sensitivity [23].

Gut dysbiosis is associated with metabolites production involved in the loss of gut homeostasis. Members of the *Sellimonas* genus, which are in higher abundance in MASLD/F+ than in MASLD/F−, are characterized by gene expression profiling involved in processes that lead to the production of short-chain fatty acid (SCFA). Despite an abundance of one of its members, *Sellimonas intestinalis,* it has been described as a candidate biomarker of gut homeostasis re-establishment [24]. This species has been observed in abundance in diverse inflammatory diseases such as rheumatoid arthritis, juvenile idiopathic arthritis, and chronic kidney disease [25,26,27]. Several studies have reported a loss of SCFAs in association with gut dysbiosis in MASLD patients, where reduced plasma SCFA levels have been associated with advanced liver fibrosis [28]. Indeed, we observed a decrease in SCFA-producing microorganisms, such as Defluviitaleaceae and Lachnospiraceae ND 3007, in our MASLD patient’s cohort. However, the gut microbiota of MASLD/F+ patients have a lower abundance of Ruminococcaceae UCG 013 compared to the MASLD/F− patients, suggesting that SCFA-producing bacteria could be involved in the fibrosis stage of the patients. Similarly, other studies also reported a lower abundance of these genera in MASLD and others, such as *Barnesiella*, *Faecalibacterium,* and *Ruminococcus* [29,30]. SCFAs are produced through the fermentation of soluble dietary fibers and can activate their free fatty acid receptors (FFARs), including G-protein coupled receptors 43 and 41 (GPR43, GPR41) [31]. The activation of these pathways inhibits immune functions in neutrophils, monocytes, and macrophages, thereby reducing the generation of inflammatory cytokines, such as the tumor necrosis factor (TNF)-α and monocyte chemotactic protein-1 [31,32]. Moreover, SCFAs are implicated in liver protection against fat deposition by inhibiting fat accumulation and promoting the metabolism of unincorporated lipids and glucose in other tissues [33]. These results suggest that the loss of SCFA-producing bacteria might favor the development of steatosis and fibrosis. In addition, SCFAs, and more particularly butyrate, regulate the intestinal barrier function by decreasing the tight junction permeability of the colonic epithelium [34]. The loss of the barrier integrity could enhance the translocation of bacterial toxins, such as the LPS and flagellin, and contribute to the systemic inflammatory response. Interestingly, LPS concentration is higher in the plasma of patients with NASH than in those with MASLD [35]. However, contrary to these findings, we also observed that various SCFA-producing bacteria, such as *Bifidobacterium*, *Prevotella*_2, *Sarcina,* and *Acidaminococcus*, positively correlated with liver stiffness. Another study also reported high concentrations of fecal SCFAs in NAFLD patients [36]. It is, therefore, necessary to investigate the role of SCFAs in MASLD progression. Studies have also identified other microbial products involved in the pathogenesis of MASLD, such as trimethylamine N-oxide (TMAO) and bacterial metabolites resulting from the metabolism of tryptophan, phenylalanine, and tyrosine [37].

In conclusion, MASLD is associated with gut dysbiosis characterized by an alteration in bacterial populations involved in bile acid metabolism and SCFA production. In addition, the presence of fibrosis is more specifically accompanied by a lower abundance of Ruminococcaceae UCG 013 and Ruminoclostridium_6 genera, and a higher abundance of *Sellimonas*, suggesting that these bacterial taxa could complement the classical predictors of MASLD severity. Future studies should aim to investigate the role of secondary bile acids and SCFAs in the pathogenesis of MASLD for the development of new therapeutic strategies.

## 4. Materials and Methods

### 4.1. Study Design

The research design was a descriptive cross-sectional case–control pilot study, conducted at the Gastroenterology Unit of the Hospital Clinico University of Chile (HCUCH), between December 2017 and February 2020. The participants included were those who consecutively attended the hepatology or general gastroenterology consultation. They were consulted regarding their willingness to participate in the study after confirming that they met the inclusion criteria. After that, the variables of interest were collected. Fecal microbiota analysis was blinded to group allocation.

### 4.2. Ethics Approval and Informed Consent

The study protocol complied with the ethical guidelines of the 1975 Declaration of Helsinki and was approved by the Scientific Ethics Committee of the HCUCH (Approval number 71, 8 November 2017, OAIC n°930/17). All subjects provided written informed consent before their enrolment.

### 4.3. Participants

Patients fulfilling the invasive and non-invasive criteria of MASLD, older than 18 years of age, were recruited from the Gastroenterology Unit of HCUCH, between the years 2017 and 2022. The diagnosis of MASLD was determined by an expert hepatologist according to AASLD Criteria [38] and considering the updated nomenclature [39]. MASLD diagnosis was based on the results of transient elastography (Fibroscan^®^) and/or biopsy histology according to the diagnostic criteria outlined by the Pathology Committee of the NASH Clinical Research Network [40,41]. These criteria defined steatosis based on the presence of lipid accumulation in more than 5% of the hepatocytes and significant fibrosis by the presence of fibrosis in the portal/periportal Zone 3 area (fibrosis stage F ≥ 2). The diagnosis of steatosis through Fibroscan^®^ was defined by a measurement of Controlled Attenuation Parameter (CAP) ≥ 294 dB/m (>S1) according to Petroff et al. [41]. The presence of significant fibrosis (F ≥ 2) was defined through Fibroscan^®^ by a stiffness ≥ 8.2 KPa according to Eddowes et al. [42].

The control group included subjects older than 18 years of age, without history of liver disease and absence of hepatic steatosis, with CAP < 294 dB/m (S0) as determined through Fibroscan^®^.

Exclusion criteria for both groups of patients and control subjects included alcohol consumption (>20 g/day), recent use of antibiotics or probiotics, history of digestive surgery (except appendectomy and cholecystectomy), uncontrolled diabetes, pregnancy, chronic diseases, chronic infection with hepatitis C or B virus, or other liver diseases, use of drugs associated with MASLD development.

Based on these considerations, the study involved the following groups:Healthy control: BMI < 25 Kg/m^2^, with no MASLD and fibrosis (Fibroscan^®^ CAP < 294 dB/m and stiffness < 8.2 pKa) (HC);Overweight/obese subjects: (BMI ≥ 25 kg/m^2^) with no MASLD and fibrosis (Fibroscan^®^ CAP < 294 dB/m and stiffness < 8.2 pKa) (Ow/Ob);Patients with MASLD (Fibroscan^®^ CAP ≥ 294 dB/m or biopsy-proven steatosis or steatohepatitis), without significant fibrosis (stiffness < 8.2 KPa, F < 2) (MASLD/F−);Patients with MASLD and significant fibrosis (Fibroscan^®^ CAP ≥ 294 dB/m and stiffness ≥ 8.2 pKa or a biopsy-proven steatosis or steatohepatitis with significant fibrosis ≥ F2) (MASLD/F+).

### 4.4. Clinical, Anthropometric, and Metabolic Variables

Age, gender, weight, height, body mass index (BMI), waist circumference (WC), comorbidities, biochemical serum parameters (cholesterol, triglycerides, alkaline phosphatase (ALP), gamma-glutamyltransferase (GGT), aspartate aminotransferase (AST), alanine aminotransferase (ALT), and albumin), alcohol (<20 g/day) and tobacco consumption, and prescribed medications were registered at recruitment. 

### 4.5. DNA Extraction, 16S rRNA Amplicon Sequencing, and Bioinformatic Analysis

Fecal samples were collected from all participants, immediately transported on ice, and stored at −80 °C until processing. DNA was extracted using DNAeasy PowerSoil^®^ Kit, Qiagen GmbH, Germantown, MD, USA according to the manufacturer’s instructions, and the concentration of eluted DNA was determined with NanoQuant. For each DNA sample, the v3–v4 region of the 16S rRNA gene was amplified, and the resulting amplicons were sequenced using the Illumina MiSeq Platform (Center for Comparative and Functional Genomics, University of Illinois, Urbana-Champaign, Urbana, IL, USA). Demultiplexed reads were imported into the DADA2 v1.22.0 pipeline [43]. Forward reads were used for downstream analysis, and primer sequences were removed from the reads using the DADA2 RemovePrimers function, followed by quality filtering and trimming using the DADA2 FilterAndTrim function. Sequences were then corrected for Illumina amplicon sequence error, dereplicated, and amplicon sequence variants (ASVs) were generated, followed by chimera removal. Taxonomic classification was performed using DADA2 to assign Taxonom and addSpecies using silva_nr_v138_st-fa-gz and silva_species_assignment_v138.fa.gz, respectively.

### 4.6. Statistical Analyses

The statistical analysis was performed and visualized using R software, version 4.1.2. Differential abundance analysis was performed using the Kruskal–Wallis test at phylum, family, and genus levels. Alpha-diversity (observed ASVs and Shannon index) and beta-diversity (Adonis) were calculated based on the ASV Table representing the relative abundances of bacterial taxa from the R package microbiome v.1.16.0 R.

## Figures and Tables

**Figure 1 ijms-25-04387-f001:**
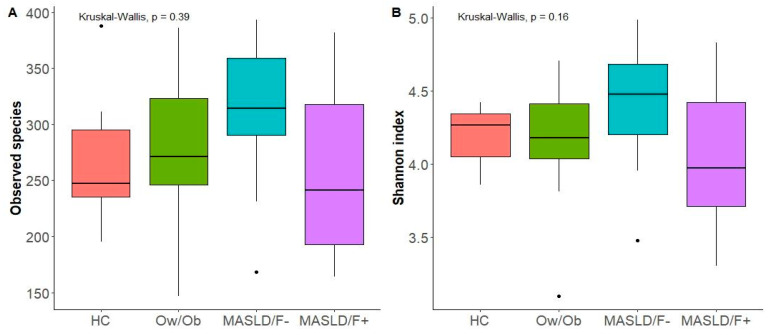
Boxplots of alpha-diversity indices. Alpha-diversity comparisons of the gut microbiota of the healthy controls (HCs), overweight/obese without MASLD (Ow/Ob), MASLD without fibrosis (MASLD/F−), and MASLD with fibrosis (MASLD/F+). (**A**) Observed species. (**B**) Shannon index. *p* < 0.05 (Kruskal–Wallis).

**Figure 2 ijms-25-04387-f002:**
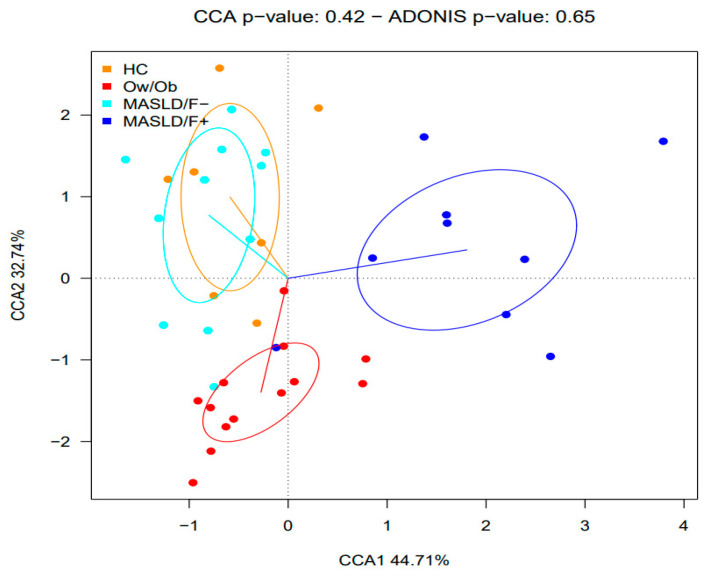
Canonical correspondence analysis (CCA) shows the correlation between the gut microbiota compositions at the species level of healthy control (HC), overweight/obese without MASLD (Ow/Ob), MASLD without fibrosis (MASLD/F−), and MASLD with fibrosis (MASLD/F+).

**Figure 3 ijms-25-04387-f003:**
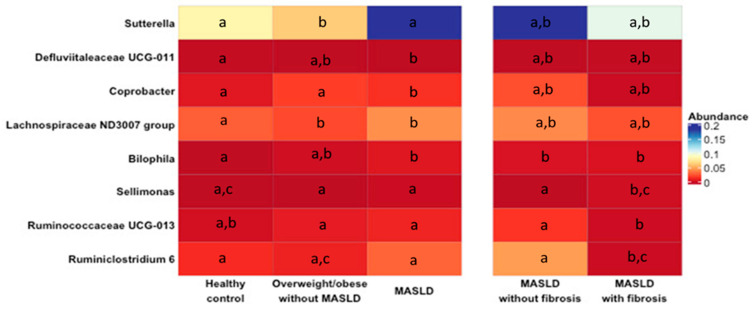
Heatmap showing the abundance difference of the gut microbiota at the genus level among healthy controls (HCs), overweight/obese without MASLD (Ow/Ob), MASLD patients, MASLD without fibrosis (MASLD/F−), and MASLD with fibrosis (MASLD/F+). For each heatmap row, values with different superscripts are significantly different (*p* < 0.05 (Kruskal–Wallis test among groups and post hoc Dunn’s test)). The scale was normalized for better visualization of the data.

**Figure 4 ijms-25-04387-f004:**
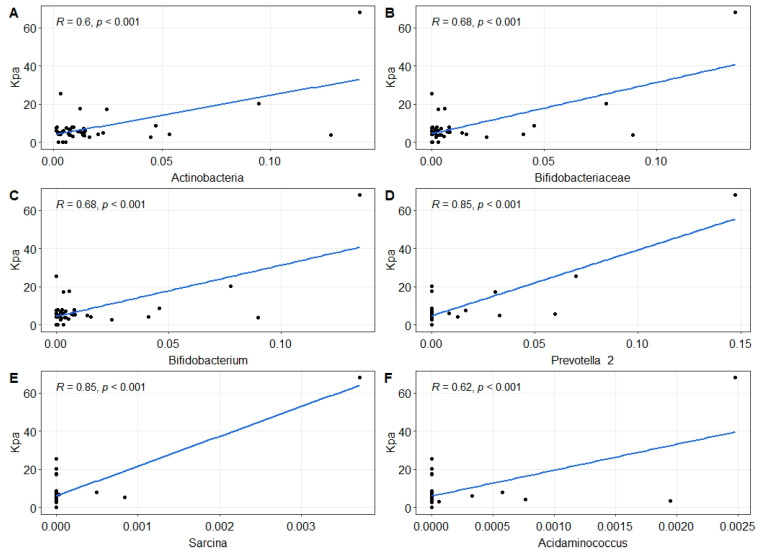
Correlations between the relative abundance of different bacterial taxa and fibrosis severity (KPa) (Pearson correlation). (**A**) Actinobacteria phylum. (**B**) Bifidobacteriaceae family. (**C**) *Bifidobacterium* genus. (**D**) *Prevotella* 2 genus. (**E**) *Sarcina* genus. (**F**) *Acidaminococcus* genus.

**Table 1 ijms-25-04387-t001:** Characteristics of the different groups of subjects/patients.

	Healthy Controls (n = 7)	Overweight/Obese Subjects with No MASLD and Fibrosis (n = 13)	Patients with MASLD and Non-Significant Fibrosis (n = 11)	Patients with MASLD and Significant Fibrosis (n = 9)	*p* Value
% women	71.4	61.5	81.8	33.3	0.1536
Age (years)	39.9 ± 17.4 (23–63)	48.9 ± 16.3 (18–73)	49.0 ± 15.1 (25–67)	63.2 ± 9.8 ^Ψ^ (47–73)	0.0293 *
Weight (Kg)	60.9 ± 6.3 (53.0–70.0)	76.2 ± 8.4 ^Ψ^ (67.0–98.0)	75.4 ± 13.8 (57.0–101-0)	83.3 ± 13.5 ^†^ (60.7–103.0)	0.0047 **
BMI (kg/m^2^)	22.3 ± 0.9 (20.8–23.7)	28.2 ± 2.7 ^†^ (25.4–34.7)	28.4 ± 3.4 ^†^ (23.4–34.1)	31.2 ± 5.1 ^ƒ^ (26.0–39.2)	0.0004 ***
Waist circumference (cm)	79.8 ± 8.8	96.4 ± 7.1	98.6 ± 9.1 ^Ψ^	101.8 ± 8.8 ^†^	0.0103 *
Nutritional status (%)					
Normal weight	100	0	18.2	0	
Overweight	0	76.9	54.5	33.3	
Obesity grade 1	0	23.1	27.3	55.6	
Obesity grade 2	0	0	0	11.1	
Biochemical parameters					
Cholesterol (mg/dL)	181.6 ± 50.6 (119–278)	193.2 ± 42.58 (125–257)	210.5 ± 63.6 (116–311)	141.6 ± 24.4 ^¥^ (103–175)	0.0247 *
Triglycerides (mg/dL)	87.1 ± 53.5 (35–180)	145.2 ± 94.3 (48–361)	154.9 ± 80.7 (69–323)	109.8 ± 83.8 (41–254)	0.1654
ALP (UI/L)	76.7 ± 19.9 (42–97)	89.44 ± 25.2 (62–121)	111.1 ± 36.3 (65–173)	124.0 ± 54.5 (67–200)	0.2163
GGT (UI/L)	17.5 ± 5.1 (13 -24)	28.2 ± 19.0 (16–76)	113 ± 150 (15–470)	177 ± 166 ^∂,†^ (35–474)	0.0031 **
AST (UI/L)	24.0 ± 5.3 (19.0–31.0)	34.3 ± 22.3 (21.0–93.0)	77.6 ± 68.6 ^Ψ^ (17.0–231.0)	47.9 ± 19.7 (21.0–83.0)	0.0227 *
ALT (UI/L)	15.2 ± 5.4 (11.0–23.0)	50.0 ± 65.5 (12.0–224.0)	121.1 ± 105.6 ^Ψ^ (14.0–289.0)	43.6 ± 23.4 (20.0–88.0)	0.0139 *
Albumin (g/dL)	4.35 ± 0.24 (4.00–4.70)	4.23 ± 0.28 (3.80–4.70)	4.62 ± 0.34 (4.20–5.20)	3.86 ± 0.45 ^§^ (3.20–4.60)	0.0078 **
Tobacco consumption (%)	1 (14.3)	3 (23.1)	1 (9.1)	1 (11.1)	
Alcohol consumption (<20 g/day) (%)	5 (71.4)	8 (61.5)	4 (36.4)	3 (33.3)	
Comorbidities N (%)					
Diabetes Mellitus	0 (0)	1 (7.7)	2 (18.2)	5 (55.5)	
Hypertension	2 (28.6)	1 (7.7)	4 (36.4)	3 (33.3)	
Dyslipidemia	2 (28.6)	4 (30.8)	3 (27.3)	2 (22.2)	
Atherosclerosis	0 (0)	0 (0)	0 (0)	0 (0)	

Data are shown as means ± SD (range). Comparison analyses were performed using Kruskal–Wallis and Chi-square tests among groups. ^Ψ^
*p* < 0.05, regarding the HC group; ^†^
*p* < 0.01, regarding the HC group; ^ƒ^
*p* < 0.001, regarding the HC group; ^∂^ *p* < 0.05, regarding overweight/obese group; ^¥^
*p* < 0.05 and ^§^
*p* < 0.01, regarding MASLD non-significant fibrosis. * *p* < 0.05, ** *p* < 0.01, *** *p* < 0.001, for different groups comparison by using analysis of variance. Abbreviations: BMI, body mass index; MASLD, metabolic dysfunction-associated steatosis liver disease; ALP: alkaline phosphatase; GGT: gamma-glutamyltransferase; AST: aspartate aminotransferase; ALT: alanine aminotransferase.

## Data Availability

The data presented in this study are available on request from the corresponding author. The data are not publicly available due to ethical reason.

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
