# Peer review of "Alteration of Gut Microbiota Composition in the Progression of Liver Damage in Patients with Metabolic Dysfunction-Associated Steatotic Liver Disease (MASLD)"

_ijms, 2024, doi:10.3390/ijms25084387_

Round 1

Reviewer 1 Report

Comments and Suggestions for Authors

This is potentially an important paper. However, the main drawback is the lack of information regarding patient selection.

The authors state” Forty subjects were recruited in the study’ without any information from where they were recruited. What was the basis for the diagnosis of MASLD? How was the diagnosis of fibrosis made?

Without clarifying this it is nearly impossible to make firm conclusions.

I suggest resubmission after this issue is addressed.

Author Response

Dear reviewer,

we welcome your suggestions and comments.

In agreement with them, we proceeded to edit the information regarding to recruit participants in this pilot study. We included information regarding where subjects were recruited, as well as non-probability sampling technique applied (please see highlighted line nº65-68; 247-249 in the manuscript; in the result and Subjects and Methods section in the manuscript, respectively). Similarly, the MASLD and fibrosis diagnosis method used is described in Abstract, Results and Subjects and Methods section, line nº 22-23; 70-71; 257-268, respectively).

Best Regards

Caroll Beltran

Reviewer 2 Report

Comments and Suggestions for Authors

The Authors analyzed the gut microbiota of 40 patients with different levels of steatotic liver disease and obesity.

Fibrosing bacterial species have been found in the gut microbiota of patients with MASLD.

The work is well structured and organized.

However, there are some things to take into consideration:

- the title says "Chilean population," but a cohort of 40 cases may not be sufficiently representative of an entire nation;

- how is the statistically significant difference in the age of the different subpopulations in Table 1 explained?

Comments on the Quality of English Language

English language is quite good

Author Response

Dear reviewer, we welcome your suggestions and comments. Attending to the recommendation, we have proceeded to erase the Chilean word in the title, as well as to specify in the abstract that this study was performed in a group of Chilean MASLD patients (please see line nº 19 in the attached manuscript). Regarding to statistic differences in the age among MASLD/F+ and HC, we proceed to specified it, by symbol, in Table 1. Similarly, we included the same specifications for weight, BMI and waist circumference. These differences can be explained by our consecutive sampling that was used in subjects’ recruitment. As results, MASLD/F+ were characterized to be older in comparison with HC group. This information agrees with previous report that have indicate an increased prevalence of MASLD, as well as advanced stage of liver fibrosis, in older patients. Then, this age difference can be a limitation of our study (mentioned in line 157; 158-161 in the attached manuscript), due to it can be considered as a modifier factor of intestinal microbiota.

Best Regards

Caroll Beltran

Round 2

Reviewer 1 Report

Comments and Suggestions for Authors

I think the authors have addressed the comments and suggest to accept after minor improvements in English.

Comments on the Quality of English Language

Minor grammatical  changes in the highlighted areas required.